# Preparation of SA–PA–LA/EG/CF CPCM and Its Application in Battery Thermal Management

**DOI:** 10.3390/nano11081902

**Published:** 2021-07-24

**Authors:** Ziqiang Liu, Juhua Huang, Ming Cao, Yafang Zhang, Jin Hu, Qiang Chen

**Affiliations:** Department of Mechanical Engineering, Nanchang University, Nanchang 330031, China; 350928919006@email.ncu.edu.cn (Z.L.); huangjuhua6@163.com (J.H.); zhangyafang1@126.com (Y.Z.); 400928918003@email.ncu.edu.cn (J.H.); 410914119004@email.ncu.edu.cn (Q.C.)

**Keywords:** composite phase change materials, battery thermal management, low-melting eutectic phase change materials, battery charging and discharging, battery temperature control

## Abstract

To improve the heat dissipation efficiency of batteries, the eutectic mass ratios of each component in the ternary low-melting phase change material (PCM), consisting of stearic acid (SA), palmitic acid (PA), and lauric acid (LA), was explored in this study. Subsequently, based on the principle of high thermal conductivity and low leakage, SA–PA–LA/expanded graphite (EG)/carbon fiber (CF) composite phase change material (CPCM) was prepared. A novel double-layer CPCM, with different melting points, was designed for the battery-temperature control test. Lastly, the thermal management performance of non-CPCM, single-layer CPCM, and double-layer CPCM was compared via multi-condition charge and discharge experiments. When the mass ratio of SA to PA is close to 8:2, better eutectic state is achieved, whereas the eutectic mass ratio of the components of SA–PA–LA in ternary PCM is 29.6:7.4:63. SA–PA–LA/EG/CF CPCM formed by physical adsorption has better mechanical properties, thermal stability, and faster heat storage and heat release rate than PCM. When the CF content in SA–PA–LA/EG/CF CPCM is 5%, and the mass ratio of SA–PA–LA to EG is 91:9, the resulting SA–PA–LA/EG/CF CPCM has lower leakage rate and better thermal conductivity. The temperature control effect of single-layer paraffin wax (PW)/EG/CF CPCM is evident when compared to the no-CPCM condition. However, the double-layer CPCM (PW/EG/CF and SA–PA–LA/EG/CF CPCM) can further reduce the temperature rise of the battery, effectively control the temperature and temperature difference, and primarily maintain the battery in a lower temperature range during usage. After adding an aluminum honeycomb to the double-layer CPCM, the double-layer CPCM exhibited better thermal conductivity and mechanical properties. Moreover, the structure showed better battery temperature control performance, while meeting the temperature control requirements during the charging and discharging cycles of the battery.

## 1. Introduction

Due to their long battery life and high power output, lithium-ion (Li-ion) batteries are considered a primary energy source for both electric and hybrid electric vehicles, with the main advantage being their high energy density [1,2]. However, a considerable amount of heat is generated in Li-ion batteries, and because of their narrow structure and close arrangement of batteries, it becomes difficult to disperse this heat, resulting in heat accumulation. These batteries have a suitable working temperature range from 20 to 45 °C, and a temperature difference of under 5 °C [3,4,5]. Large deviations from the ambient temperature affect the performance and service life of the battery. When a Li-ion battery is charged or discharged rapidly, the internal temperature of the battery sharply rises above the ambient range, which becomes extremely unfavorable for its long-term use, and may even lead to thermal runaway of the battery [6]. Therefore, in order to preserve the performance and service life of the battery, it is necessary to take effective battery thermal management measures.

An efficient battery thermal management scheme can keep the temperature and temperature difference of the battery under control [7]. As a high latent heat energy storage material, PCM has the advantage of a simple structure, with no additional energy consumption during the process of battery heat dissipation, when compared with air-cooled or liquid-cooled battery thermal management schemes [8,9]. However, PCM itself possesses many defects, such as easy flow and low thermal conductivity after melting. CPCM can be prepared by mixing PCM and a PCM carrier. Due to the loose and porous structure of the EG, CPCM prepared with EG as a carrier material has better overall performance [10]. Zhang et al. used nine heat-conducting media, including diatomite, organic bentonite, and EG, to improve the thermal conductivity of PCM, and found that EG had the best overall effect [11]. Jiang et al. explored the optimal ratio of PW to EG in CPCM while searching for the best material for battery heat dissipation, and found that CPCM had great heat conduction and leakage prevention properties when the EG content was 16%–20% [12]. Ling compared the thermal management performance of silica/RT44HC and EG/RT44HC CPCMs in Li-ion battery packs, and found that the latter reduced the temperature difference between batteries more effectively [13].

PCM is the main component in CPCM. Generally, PCM is divided into two categories: inorganic and organic. Inorganic PCM is mainly inorganic hydrated salt, whereas the most commonly used organic PCM is PW [14,15]. There are many studies on CPCM that utilize PW as PCM. Li et al. found that PW can effectively penetrate a porous ceramic carrier after melting, and the prepared PW/porous ceramic CPCM possessed great mechanical strength and latent heat properties [16]. Gulfam et al. listed several advanced applications based on PW CPCM in their review [17]. Apart from PW, fatty acids are another type of phase change energy storage material with potential applications. Different fatty acids can be melted and mixed together to form a eutectic mixture, which can effectively reduce the phase transition temperature of fatty acids, thereby broadening the resulting PCM’s phase transition temperature range, thus making them more useful [18,19,20]. Kahwaji et al. prepared a capric acid/tetradecanoic acid eutectic mixture, and studied its thermal properties and thermal reliability. They found that the lowest melting point of the eutectic mixture was 20.5 °C [21]. Alva et al. characterized a mixture of palmitic acid (PA)/myristic acid (MA), and found that the melting temperature and latent heat of MA–PA PCM at their eutectic point were 45.48 °C and 176.12 kJ/kg, respectively [22]. Zhang found that when the mass ratio of lauric acid to palmitic acid was 22.5:77.5, the melting temperature of PCM was the lowest [23]. Carolina prepared three eutectic mixtures by mixing capric acid, lauric acid, and palmitic acid, and found that low-melting point eutectic PCM is suitable for applications in various industries such as construction, food, household hot water tanks, etc. [24]. Two types of fatty acids are mixed to form binary eutectic mixtures, whereas three kinds of fatty acids can be mixed to prepare ternary eutectic mixtures. Currently, the research on ternary eutectic fatty acids, and the application of multicomponent eutectic fatty acids in battery thermal management, is limited.

In this study, PW/EG CPCM and a new SA–PA–LA/EG CPCM were prepared, with EG as the carrier material. The CF material, having superior thermal conductivity, was added to the CPCM to enhance its thermal conductivity. In addition, PW/EG/CF and SA–PA–LA/EG/CF were combined to form a double-layer structure having both a high- and low-melting point CPCM temperature control system. In order to further improve the thermal conductivity and mechanical strength of the double-layer CPCM, this study also tried to press the CPCM into the aluminum honeycomb to obtain a double-layer aluminum honeycomb structure. This double-layer CPCM was pasted on the surface of the battery for charging and discharging experiments. Finally, the temperature control performance of the CPCM, with different discharge rates of the battery, was analyzed through the temperature test data.

## 2. Experiment Methodology

### 2.1. Preparation of CPCMs

#### 2.1.1. Experimental Materials

This study utilized two CPCMs, namely, PW/EG/CF and SA–PA–LA/EG/CF CPCM. The PW chosen was a high-purity PW from Hangzhou Ruer New Energy Technology Co., Ltd., Hangzhou, China, with a melting point of 42–45 °C. Fifty-mesh EG was used as the PCM carrier, which was provided by the Qingdao Huatai Graphite Company, Qingdao, China. Fifty-mesh CF was used as a thermally conductive filler, which was acquired from the Toray Company, Tokyo, Japan. SA (C_18_H_36_O_2_) used in this study had a melting point of 54–57 °C and a molar mass of 284 g/mol; PA (C_16_H_32_O_2_) had a melting point of 63–64 °C and a molar mass of 256 g/mol; and LA (C_12_H_24_O_2_) had a melting point of 44–48 °C and a molar mass of 200 g/mol. All three fatty acids were purchased from Shanghai Lin En Technology Development Co., Ltd., Shanghai, China. The thickness of the aluminum honeycomb panel was 5 mm, the wall thickness was 0.4 mm, and the side length was 4 mm. It was purchased from Hebei Kaicheng Metal Products Co., Ltd., Hebei, China.

#### 2.1.2. Preparation Methods

The preparation process of SA–PA–LA/EG/CF CPCM is complex. Low-melting point SA–PA–LA eutectic PCM needs to be prepared before preparing the CPCM. First of all, SA, and the corresponding amount of PA, were added into a beaker in mass ratios of 0%, 20%, 40%, 60%, 80%, and 100%, for melting. The beaker was bathed in water at 80 °C, and the fatty acid mixture was stirred with a glass rod. After the mixture became uniform, the beaker was quickly removed and placed in a low-temperature (10 °C) experiment box. A temperature detector was used to record the temperature of the mixture, and a step cooling curve was plotted from these data. The proportion of binary fatty acids (SA–PA) with the lowest phase transformation freezing point was selected on the basis of the step cooling curve, and the eutectic point of the two fatty acids was determined by taking mass ratios around that point (the solidification and melting temperatures of the polybasic fatty acids were the lowest at the eutectic point). For example, when the mass ratio of SA was 80%, the freezing point of the binary fatty acid was the lowest, and its eutectic point was determined from mass ratios around the 80% mark (71%, 74%, 77%, 80%, 83%, 86%, and 89%). After finding the mass ratio of the eutectic point of SA–PA, the binary fatty acid with this ratio was taken as a single new fatty acid, and mixed with LA by melting. Similarly, a more accurate mass ratio of ternary eutectic PCMs (SA–PA–LA) was also found.

After preparing the ternary eutectic PCM, the CPCM was prepared according to the preparation method shown in Figure 1. The PCM was melted in a water bath and Antioxidant 1010 powder was added. After stirring for 3 min, EG was added, and it was stirred for another 10 min, until the PCM was completely absorbed by EG. Finally, CF was added to the mix, and stirred to obtain SA–PA–LA/EG/CF CPCM. The uncooked and unformed CPCM was quickly poured into a mold, and the mold was covered and hot-pressed using a tablet press. The length and width of the prepared CPCM block were consistent with the size of the battery. The CPCM block contained single-layer and double-layer structures. The thickness of the single-layer CPCM was 10 mm, and the double-layer CPCM was composed of two 5 mm CPCM blocks. Among them, the double-layer CPCM included two structures, one with aluminum honeycomb and one without aluminum honeycomb. The preparation method of PW/EG/CF CPCM was almost similar to that of SA–PA–LA/EG/CF CPCM. The main difference is that the ternary eutectic PCM was replaced with PW during the preparation process.

### 2.2. Characterization of CPCMs

After preparation of the CPCMs, a variety of characterization tests were carried out in this study. A scanning electron microscope (SEM, FEI Quanta200F, Waltham, MA, USA) was used to observe the micro morphology of EG, PCM/EG, and PCM/EG/CF at room temperature. X-ray diffraction (XRD, D8 ADVANCE Bruker, Germany Karl Sruhe) was used to observe the crystal structure of the material kept in a nitrogen-filled environment, at a test angle of 10–90°. A Fourier-transform infrared spectrometer (FTIR, Thermo Fisher Scientific, Waltham, MA, USA) was employed to observe the chemical changes between the components of the CPCM. The wavelength of light used in the test was 2.5–25 µm, and the wavenumber range was 500–3500 cm^−1^. A thermogravimetric analyzer (TGA4000, PE, Waltham, MA, USA) was utilized to test the thermal stability of the material. The test temperature range was kept at 0–600 °C, and the heating rate was 10 °C/min. A differential scanning calorimeter (DSC8000, PE, Waltham, MA, USA) was used to study the latent heat of the phase change process of PCM. The temperature range in the test was 10–90 °C, and the heating and heat release rates were both 5 °C/min. Under the control of a microcomputer, an electronic universal testing machine (WDW-100E, Jinan, China) was employed to measure the compressive strength and durability of the rectangular single-layer and double-layer samples. The accuracy of the measurement was ±0.5%. A hot-wire method thermal conductivity meter (TC3000, XIATECH, Xi’an, China) was used to measure the thermal conductivity of the sample. A temperature data acquisition instrument (MR8875–30, HIOKI, Nagano, Japan) was utilized to record the step cooling curve of the low-melting point eutectic PCM. A high- and low-temperature test box (KW-890, KOWINTEST, Dongguan, China) was used to conduct a rapid heat absorption and heat release performance test on the samples. Notably, the DFT theory developed by Rivelino et al. [25,26], i.e., the synthetic growth method, is effective for the in-depth analysis of material stability and possible chemical reactions, and it has been applied to complex systems. In addition, although PW/EG/CF and SA–PA–LA/EG/CF CPCMs were involved in this study, the author has performed the application and material characterization of CPCM based on PW/EG many times in previous studies [27,28,29]; therefore, the characterization test materials involved in this study mainly included the SA–PA–LA ternary eutectic PCM and SA–PA–LA/EG/CF CPCM.

### 2.3. Battery Charge and Discharge Experiment

To explore the temperature-control characteristics of CPCMs on batteries, this study conducted a charge and discharge experiment on a battery, as shown in Figure 2. Figure 2b shows an optical diagram of the battery module. The molded CPCM blocks were tightly attached to the surface, on both sides of the battery, and the two tabs of the battery were connected to two charging and discharging ports. Figure 2a shows a schematic of the battery charging and discharging system. Two temperature sensors were placed between the battery and the CPCM as monitoring points (points 1 and 2 in Figure 2). Point 1 is the center of the front surface of the battery, and point 2 is fixed at the edge of the battery surface. The connected battery module was placed in a temperature test box at 25 °C to simulate a room temperature environment. The battery was then charged and discharged by the battery charging and discharging equipment. The maximum discharge rate of the battery was 2C; therefore, only 1C and 2C discharges were carried out in the experiment. The temperature of the battery surface was transmitted to the data acquisition system through the sensor, and the temperature data were retrieved by the computer, where the temperature response curve was plotted. The battery possessed temperature-control systems in all three types of samples: no-CPCM, single-layer CPCM, and double-layer CPCM (low melting point fatty acid on the inner layer, high melting point PW on the outer layer). The battery was purchased from Dongguan Xingfeng Lithium Battery Technology Company (Dongguan, China), and the battery parameters are shown in Table 1. The battery charging and discharging equipment was purchased from Shenzhen Hengyi Energy Technology Co., Ltd. (model number: PT120300A, Shenzhen, China). The temperature data acquisition equipment was acquired from a Japanese company named HIOKI (model number: MR8902, Nagano, Japan). The high- and low-temperature test chambers were produced by Honghe Laboratory Equipment Co., Ltd. (model number: WGD-0208, Shanghai, China).

## 3. Results and Discussion

### 3.1. Binary and Ternary Eutectic PCMs

#### 3.1.1. Step Cooling Characteristics of Binary PCMs

The SA–PA mixture was cooled in a low-temperature test chamber, and the temperature change in binary PCM, having different proportions, was monitored by the data acquisition instrument. Figure 3 shows the resulting step cooling curve. Solidification reactions of SA, PA, and SA–PA occurred during the cooling process, while the temperature drop rate of fatty acids in the solidification stage was relatively slow. In Figure 3a, the mass ratio of SA was taken in 20% increments, and the step cooling curves of binary PCMs are different. SA and PA begin to solidify around 53.5 °C and 57 °C, respectively. From the curve, when the mass ratio of SA was 80%, the solidification temperature of binary PCM was the lowest, with the lowest point being 50.5 °C. In order to find a more accurate SA–PA eutectic point, SA mass ratios around 80% were tested. The step cooling curves of binary PCMs with SA mass ratios of 71% to 89%, in 3% increments, were plotted (Figure 3b). It was found that the solidification temperature was still the lowest when the mass ratio of SA was exactly 80%. This indicated that SA–PA reached the eutectic state when the ratio of SA-to-PA was close to 8:2.

#### 3.1.2. Phase Change Properties of Binary PCMs

Figure 4 shows the DSC spectra of SA, PA, and SA–PA (8:2). In the figure, the negative peak represent the endothermic melting process of PCM, whereas the positive peak represent the exothermic solidification process. The peak melting temperature and the latent heat of phase change of SA were 57 °C and 176.59 J/g, respectively, whereas those of PA were 64 °C and 203.99 J/g, respectively. The endothermic and exothermic peak temperatures of SA–PA were lower than the temperatures of the two corresponding fatty acids. Additionally, there was only one endothermic and exothermic peak during the heating and cooling process of SA–PA. When the mass ratio of SA to PA was 8:2, SA–PA was in a better low-melting and eutectic state. In addition, all three materials represented in the figure had a certain degree of undercooling, which had an adverse effect on the heat exchange efficiency of the PCM. However, when compared with SA and PA, the transverse temperature span of the endothermic and exothermic peaks of SA–PA is a lot lower, which indicates that the phase transformation process of eutectic SA–PA is faster than that of SA and PA. This suggests that eutectic PCMs can effectively improve the thermal management efficiency. Although the melting point of SA–PA was lower than that of SA and PA, the peak melting temperature of SA–PA (56 °C) was still much higher than the suitable upper limit of 45 °C. It was necessary to prepare ternary eutectic PCMs to further reduce the melting point.

#### 3.1.3. Step Cooling Characteristics of Ternary PCMs

After the optimum mass ratio of the binary eutectic PCM was obtained, SA–PA was regarded as a whole and mixed with LA to prepare the ternary PCM. From the step cooling curve (Figure 5a), the temperature changes in all the samples during the solidification process were very slow. The start temperature was kept at 32 °C, and the resulting solidification temperatures of the six samples are clearly very distinct. The highest solidification temperature was obtained when the mass ratio of SA–PA to SA–PA–LA was 100%; the lowest solidification temperature was obtained when the mass ratio of SA–PA to SA–PA–LA was 40%. To find an accurate eutectic point of the ternary PCM, mass ratios around 40% were tested (Figure 5b). According to the data, the SA–PA–LA ternary PCM reached eutectic state when the proportion of SA–PA was close to 37%. At this point, the solidification temperature of the material was 30.5 °C. 

#### 3.1.4. Phase Change Properties of Ternary PCMs

Figure 6 shows the DSC spectrum of the ternary eutectic PCM and its constituents. The proportion of SA–PA was 37%, that of LA was 63%, while the mass ratio of SA to PA was 8:2. Figure 6a shows that the melting and freezing points of SA–PA–LA are lower than those of SA–PA and LA. Moreover, SA–PA–LA only had one endothermic and exothermic peak, indicating that the ternary PCM was in a low-melting and eutectic state. In addition, the melting peak temperatures of SA–PA, LA, and SA–PA–LA were 56 °C, 44.5 °C, and 37.5 °C, respectively, and their latent phase change heats were 177.27, 186.34, and 160 J/g, respectively. Figure 6b shows the data obtained after multiple cycles of the SA–PA–LA DSC test. The DSC thermal stability of the ternary eutectic PCM was good, and the latent phase change heat and the peak temperature had not changed significantly. This shows that the prepared ternary eutectic PCM can be recycled for practical applications.

Figure 7 shows an experimental equilibrium phase diagram, drawn based on the step cooling curve. In the figure, as the mass fraction of each component changes, the average solidification temperature of the PCM also changes significantly. Figure 7a shows that the solidification temperatures of SA and PA are 53.5 °C and 57 °C, respectively, whereas the minimum solidification temperature of SA–PA binary PCM is approximately 50.5 °C. As shown in Figure 7b, the freezing temperature of LA was 42.5 °C, whereas that of SA–PA–LA was as low as 28.5 °C. From the data trends in the two experimental phase diagrams, it can be seen that the solidification temperatures of the binary PCM and the ternary PCM decreased first and then increased with changes in the mass fraction of each component. The results show that both SA–PA and SA–PA–LA in this study must have had low-melting eutectic points. The eutectic mass ratio of SA–PA was close to 8:2, whereas that of SA–PA and LA in ternary PCM was close to 37:63, and that of each component in SA–PA–LA was 29.6:7.4:63.

### 3.2. CPCMs

#### 3.2.1. Microscopic Characterization

Based on the CPCM preparation process shown in Figure 1, EG was added to the prepared ternary eutectic PCM as a carrier, and CF as a thermally conductive filler. The SEM image shows the micro-combination mechanism of each component in CPCM. Figure 8 shows the SEM images of EG, SA–PA–LA/EG, and SA–PA–LA/EG/CF. Figure 8a shows an SEM image of the 50 mesh EG, enlarged by 400 times. It can be seen as a long strip or with a worm-like microscopic morphology, and there are many cracks and voids on its surface. Figure 8b is a microscopic image of SA–PA–LA/EG CPCM, with 12,000 times magnification. A large amount of PCM had been adsorbed in the cracks present on the EG surface. Figure 8c shows the SEM image of SA–PA–LA/EG mixed with a small amount of CF. The CF has a long strip shape, and it became more interlaced as more of it was added.

#### 3.2.2. Leakage and Thermal Conductivity Test

Leakage rate is a critical parameter that affects the recycling of CPCM. Leakage of PCM weakens the heat absorption capacity of CPCM. Generally, the higher the content of EG, the lower the leakage rate of CPCM. However, an increase in the EG content means a decrease in the PCM content, which will lead to decreased latent heat of CPCM. Figure 9 shows the leakage of SA–PA–LA/EG CPCM samples with different contents of EG. All the samples shown in the figure were baked in an oven at 80 °C for 5 h. Evidently, the highest amount of PCM leakage was observed in the sample with 4% EG content. However, no evident leakage was seen in the samples with 9% or more EG content. Therefore, at 9% EG content, the CPCM not only ensured a low leakage rate, but also minimized the EG content, thus making sure of a sufficiently high latent phase change heat.

Ensuring that SA–PA–LA/EG/CF CPCM has low leakage, high thermal conductivity, and a better composition quality ratio is one of the fundamental challenges in the preparation process of CPCM. In SA–PA–LA/EG/CF, because EG is a PCM carrier, first, the EG content was adjusted according to the data obtained from Figure 9, to solve the leakage problem of SA–PA–LA/EG. Next, an appropriate proportion of CF was added to SA–PA–LA/EG to improve the thermal conductivity of the resulting SA–PA–LA/EG/CF. Figure 10a compares the leakage rate of SA–PA–LA/EG, with different EG contents. When the EG content was 4%, the leakage rate of CPCM was as high as 13.4%. That rate dropped to just 0.5% when the EG content was 9%. Therefore, in this study, the EG content was locked at 9% when synthesizing SA–PA–LA/EG, and CF was further added in different proportions to form the CPCM. Figure 10b shows the change in thermal conductivity of SA–PA–LA/EG/CF with the change in CF content. The thermal conductivity of CPCM is anisotropic; therefore, the axial and radial thermal conductivity of the samples were measured separately. Figure 10 shows that the thermal conductivity of CPCM increased with the CF content. However, the greatest increase in axial and radial thermal conductivity is observed when the CF content was between 4% and 5%. After the 5% mark, the increase in thermal conductivity slowed down. Hence, when the mass fraction of CF was 5%, a heat conduction network may be formed inside the CPCM. This proved that at 5% CF content in SA–PA–LA/EG/CF CPCM, and at a 91:9 mass ratio of SA–PA–LA to EG, the resulting SA–PA–LA/EG/CF CPCM possessed both lower leakage rate and good thermal conductivity. Figure 10c shows the thermal conductivity data of SA–PA–LA/EG/CF CPCM with the change in the CF content after adding the aluminum honeycomb structure. When an aluminum honeycomb was added, the axial thermal conductivity of the CPCM was higher than that of radial thermal conductivity, which was the opposite to the result in Figure 10b. This phenomenon may be attributed to the special 3D structure of the aluminum honeycomb, and the high axial thermal conductivity has significant advantages for batteries that mainly dissipate heat in the axial direction. In addition, the thermal conductivity of the CPCM was significantly improved after adding the aluminum honeycomb, and the thermal conductivity of most of the samples was greater than 5 W·m^−1^·K^−1^, which can be attributed to the high thermal conductivity of aluminum.

#### 3.2.3. Thermal Stability Study

To explore the thermal stability of CPCM, TGA tests were carried out on SA–PA–LA, SA–PA–LA/EG, and SA–PA–LA/EG/CF samples. Figure 11 shows the results. The mass fraction of EG in SA–PA–LA/EG CPCM was 9%, whereas that of CF in SA–PA–LA/EG/CF CPCM was 5%, and the mass ratio of SA–PA–LA to EG was 91:9. It can be seen from the figure that the SA–PA–LA eutectic PCM began to decompose at 103 °C, and the mass fraction became 0 after 230 °C. SA–PA–LA/EG and SA–PA–LA/EG/CF CPCMs began to degrade at 95.4 °C and 90.6 °C, respectively; however, the mass fractions of the two materials remained unchanged after 205 °C, with the remaining masses being 9.13% and 13.05%, respectively. The above results indicate that ternary eutectic PCM can be completely decomposed in the test, whereas EG and CF will not decompose in the range of 0–700 °C. From the chemical nature of the substance, fatty acids contain longer carbon chains, which are easily decomposed when heated. In addition, polybasic fatty acids are compounds whose thermal stability is lower than that of pure fatty acids. However, CF and EG mainly contain C—C covalent bonds and exist in a hexagonal structure, giving them a better thermal stability. From the initial decomposition temperature of each sample, it can be seen that SA–PA–LA/EG/CF CPCM decomposed first, whereas SA–PA–LA decomposed the slowest. This may be due to the better thermal conductivity of CPCMs, which accelerates the decomposition rate of PCMs in them. Additionally, the decomposition temperatures of all three materials were higher than 90 °C; hence, the thermal stability of SA–PA–LA and its CPCM is desirable under general environmental conditions.

#### 3.2.4. Infrared Spectrum Analysis

Figure 12 shows the FTIR spectra of the SA–PA–LA/EG/CF CPCM and its components. The FTIR images of EG and CF are similar, and there are two obvious characteristic absorption peaks at 1600 cm^−1^ and 2370 cm^−1^, which correspond to the characteristic absorption peaks of H–O–H and O=C=O, respectively. In the FTIR curve of SA–PA–LA, the characteristic absorption peak at 2921 cm^−1^ is caused by the stretching vibration of —CH_3_ and —CH_2_. Similarly, the characteristic absorption peak at 1715 cm^−1^ is caused by the stretching vibration of C=O, the one at 1468 cm^−1^ is due to the bending vibration of —CH_2_, whereas those at 1290, 936, and 725 cm^−1^ are caused by C—O and C—H. The characteristic absorption peaks of SA–PA–LA/EG and SA–PA–LA/EG/CF are similar; both have characteristic absorption peaks at 725, 936, 1290, 1468, 1600, 1715, 2370, and 2921 cm^−1^. Comparing the FTIR curves of the five samples, it can be observed that the peaks of CPCM are the superposition of the peaks of SA–PA–LA, EG, and CF, and there are no evident increases or decreases in the absorption peaks. The recombination of SA–PA–LA, EG, and CF did not include any chemical reactions; only physical adsorption occurred. The aforementioned FTIR analysis data based on the polybasic fatty acid CPCM are consistent with previously reported theoretical results [30,31,32].

#### 3.2.5. Heat Storage and Heat Release Performance

The energy storage and release rate of CPCM are important factors affecting its application. Figure 13 shows the temperature change curves of the storage and heat release of three samples. As shown, in the heat storage process, it took 40 min for the SA–PA–LA to rise from 15 °C to 50 °C, whereas it only took 23.5 min for SA–PA–LA/EG/CF to do the same. Thus, the temperature-rise time is reduced by 41%, as compared to SA–PA–LA. The solid–liquid phase transition occurred between 34.5°C and 38.5 °C for both samples. In this stage, SA–PA–LA took 20 min, whereas SA–PA–LA/EG/CF took only 7 min. In the exothermic stage, it took 68 min and 41 min for SA–PA–LA and SA–PA–LA/EG/CF to decrease from 50 °C to 15 °C, respectively. Additionally, it took 40 min and 17 min for the liquid–solid phase transformations of SA–PA–LA and SA–PA–LA/EG/CF, respectively. The results show that the heat storage and heat release rate of the SA–PA–LA/EG/CF CPCM are superior to those of SA–PA–LA. This is because the high thermal conductivity of EG and CF promotes the heat storage and heat release ability of PCM. When SA–PA–LA/EG/CF was pressed into the aluminum honeycomb, the heat storage and release rates of the CPCM were further accelerated. It took 19 min for CPCM to rise from 15 °C to 50 °C, and the solid–liquid phase transition time was only 5 min. In the exothermic stage, the CPCM with an aluminum honeycomb took 27 min, and the liquid-to-solid phase transformation process only took 11 min. The results show that the addition of aluminum honeycomb promoted the heat storage and heat release rate of SA–PA–LA/EG/CF.

#### 3.2.6. Mechanical Property Test

CPCM should have a certain degree of compression resistance and anti-destructive ability during its application. Figure 14a,b show the stress test results of the CPCM without the aluminum honeycomb, before and after the electronic universal testing machine. The size of the test sample was 50 × 70 × 10 mm^3^. After compression, cracks appeared on the surface of the single-layer and double-layer CPCM blocks. However, overall, the surface of the CPCM block was smooth, and there was no evident chipping off. This smooth surface ensured that the battery and the CPCMs were closely bonded. Figure 14c shows the compression deformation diagram of the double-layer CPCM with the aluminum honeycomb. The CPCM only bent and did not fracture. Figure 14d shows the stress–strain curves of the above three samples. In the figure, the deformation of the CPCM gradually increased with the increase in the stress. The single-layer CPCM block started to fracture at 37.7 kPa, whereas the double-layer CPCM without an aluminum honeycomb started to fracture at 24.5 kPa, and the CPCM with an aluminum honeycomb did not fracture, even at 50 kPa. In addition, under the same stress conditions, the deformation of the double-layer CPCM was smaller than that of the single-layer CPCM, and the aluminum honeycomb further strengthened the mechanical properties of the double-layer CPCM.

### 3.3. Temperature Response Curve of the Battery

#### 3.3.1. Heat Dissipation without CPCM

The 1C and 2C discharge experiments on the single Li-ion battery were carried out based on the systems shown in Figure 2a. Figure 15a shows the temperature rise curve of point 1 and point 2 on the surface of the battery without CPCM. As shown, the surface temperature of the battery increased continuously with the discharge time, and the higher the discharge rate, the faster the temperature rise of the battery. When the 1C discharge ended, the temperatures at point 1 and 2 were 58.88 °C and 55.77 °C, respectively. When the 2C discharge ended, the temperatures at points 1 and 2 were 73.77 °C and 62.96 °C, respectively. Notably, the temperature at the center of the battery surface (point 1) was always greater than that at the edge of the battery (point 2). In addition, the temperature of the battery increased greatly during the discharging process, and exceeded the appropriate upper limit of 45 °C. Figure 15b shows the temperature difference curve between point 1 and point 2, under 1C and 2C discharge rates. The maximum temperature difference during the 1C discharge was 4.47 °C, whereas that during the 2C discharge was 10.81 °C, which considerably exceeded the maximum appropriate temperature difference of 5 °C. Therefore, the maximum temperature and temperature difference of the battery, under the condition of not taking heat dissipation measures, exceeded the appropriate maximum values, which was unsuitable for long-term use of the battery.

#### 3.3.2. Single-Layer CPCM Heat Dissipation

Figure 16a shows the battery discharge temperature-rise curve while using the PW/EG/CF CPCM. The thickness of the CPCM was 10 mm, and its length and width were the same as the battery size. In Figure 16, the temperatures at points 1 and 2, at the end of the 1C discharge, were 41.94 °C and 41.27 °C, respectively, and their temperatures at the end of the 2C discharge were 43.86 °C and 43.13 °C, respectively. Figure 16b shows that the maximum temperature difference between points 1 and 2 during 1C and 2C discharge was 1.59 °C and 1.33 °C, respectively. In addition, the maximum temperature difference did not appear at the end of the discharge; instead, it shifted to the initial melting stage of the PCM. Compared with the case with no heat dissipation measures, the temperature rise of the battery was significantly reduced after the CPCM was used as a thermal management measure, and the maximum temperature and temperature difference of the battery were also within the appropriate range.

#### 3.3.3. Double-Layer CPCM Heat Dissipation

Figure 17a shows the battery temperature rise curve in the SA–PA–LA/EG/CF and PW/EG/CF double-layer structure (Figure 2b) for heat dissipation. The thickness of both CPCMs was 5 mm, and the SA–PA–LA/EG/CF CPCM was the inner layer attached to the battery surface. In Figure 17a, the temperatures of points 1 and 2, after the 1C discharge, were 36.53 °C and 35.75 °C, respectively, and those after the 2C discharge were 43.22 °C and 42.89 °C, respectively. There was only one light gradient in the temperature curve of the monitoring point during the 1C discharge, indicating that only a part of the inner SA–PA–LA/EG/CF CPCM was melted in the process. Compared with the single-layer heat dissipation curve (Figure 16), the temperature rise of the double-layer CPCM with different melting points under the 1C discharge conditions was significantly improved, and the temperatures at points 1 and 2 were reduced by 5.41 °C and 5.52 °C, respectively. Under the 2C discharge rate, the temperatures at points 1 and 2 were lowered by 0.76 °C and 0.36 °C, respectively. Mostly, the battery temperature under this condition was in the low range of 30–40 °C, whereas this range was 40–45 °C in the case of single-layer CPCM. In addition, two phase transition gradients appeared in the battery temperature rise curve during the 2C discharge, which indicated that the SA–PA–LA/EG/CF CPCM in the inner layer had completely melted, whereas the PW/EG/CF CPCM in the outer layer has just begun to melt. Figure 17b shows the temperature difference curve between points 1 and 2 under 1C and 2C conditions. As shown, the maximum temperature differences during 1C and 2C discharge were 0.78 °C and 1.47 °C, respectively, which is in the optimal range of under 5 °C. From the above results, it can be concluded that the temperature control method of the double-layer CPCM can better meet the temperature control requirements (temperature, temperature rise, and temperature difference) of the battery at different discharge rates.

Figure 18 shows the battery temperature response characteristic curve when using a double-layer aluminum honeycomb structure for heat dissipation. The CPCMs in the two-layer aluminum honeycomb were SA–PA–LA/EG/CF and PW/EG/CF. From the data shown in Figure 18a,b, at the end of the 1C discharge, the temperatures at the two monitoring points were 35.06 °C and 35.00 °C. At the end of 2C discharge, the temperatures at the two monitoring points were 42.55 °C and 42.38 °C. The maximum temperature differences between the monitoring points during 1C and 2C discharges were 0.34 °C and 0.18 °C, respectively. Compared with the temperature control effect of the double-layer CPCM without the aluminum honeycomb, the double-layer CPCM with the aluminum honeycomb slightly improved the temperature and temperature difference of the battery. Figure 18c shows the cyclic charge–discharge temperature control test of 1C_discharge_-0.5C_charge_ and 2C_discharge_-0.5C_charge_ on the CPCM with the double-layer aluminum honeycomb structure. The data indicate that the temperature change law between the monitoring points is consistent and that the battery temperature at the end of each charge or discharge cycle is higher than the temperature at the end of the previous charge or discharge cycle. After three charging and discharging cycles of 1C and 2C, the surface temperature of the battery was still controlled within a suitable temperature range. This result shows that the CPCM with the double-layer aluminum honeycomb structure can effectively meet the temperature control requirements during the charging and discharging cycles of the battery.

## 4. Conclusions

In this experiment, the eutectic mass ratio of the SA–PA–LA ternary low-melting point PCM was found, and SA–PA–LA/EG/CF CPCM, with both a low leakage rate and high thermal conductivity, was prepared. Subsequently, the temperature control results of no PCM, single-layer CPCM, double-layer CPCM and double-layer aluminum honeycomb structure CPCM were compared through charge and discharge experiments. After comprehensive material characterization results and battery temperature control data, the following conclusions can be made.

The preparation of SA–PA–LA ternary eutectic PCM is carried out in two steps. First, the eutectic ratio of SA and PA is determined, and then the eutectic mass ratio of SA–PA and LA is explored. From the step cooling curve, the eutectic mass ratio of SA to PA is close to 8:2, and the eutectic mass ratio of SA–PA to LA is close to 37:63. Through the DSC test of PCMs, it was found that SA–PA and SA–PA–LA only have one absorption and exothermic peak during the heating and cooling processes, and the melting and solidification temperatures of these two PCMs are significantly reduced. This indicates that the above samples are in a good eutectic and low-melting point state. From the equilibrium phase diagram, it can be seen that the solidification temperature of PCM first decreases, and then increases with the mass fraction of each component. The minimum solidification temperature of SA–PA binary PCM is 50.5 °C, whereas that of SA–PA–LA drops as low as 28.5 °C.

When preparing CPCM, the ratio of PCM to EG should be adjusted to control the leakage rate, and the thermal conductivity of the material is increased by changing the ratio of PCM/EG to CF, thereby preparing CPCM with both thermal conductivity and low leakage. The results of leakage rate, thermal conductivity test, and SEM showed that EG can adsorb the melted PCM well. When the EG content was 9%, there was no substantial leakage of CPCM, and when the CF content was between 4% and 5%, the axial and radial thermal conductivity of CPCM increased significantly. The decomposition temperatures of SA–PA–LA, SA–PA–LA/EG, and SA–PA–LA/EG/CF were higher than 90 °C; therefore, they have good thermal stability under general environmental conditions. The FTIR curves showed that there were no chemical reactions between SA–PA–LA, EG, and CF, only physical adsorption. In the rapid heat accumulation and heat release experiments, the solid–liquid transition stages of SA–PA–LA and SA–PA–LA/EG/CF took 20 min and 7 min, respectively, whereas their liquid–solid phase transitions took 40 min and 17 min, respectively. The heat storage and heat release rate of SA–PA–LA/EG/CF were much higher than that of SA–PA–LA, which is conducive to strengthening the heat exchange capacity of CPCM to the battery. The single-layer and double-layer CPCMs have certain stress resistance, and there is no evident debris shedding after compression. The CPCM with an aluminum honeycomb structure has better thermal conductivity and resistance to stress and strain, and also has higher heat storage and heat release rates.

According to the discharge experiment of the Li-ion battery, its surface temperature increases continuously with discharge time, and the higher the discharge rate, the faster the temperature rise of the battery. When CPCM was not used for heat dissipation, the temperature-rise range of the battery monitoring point was large, and the temperature and temperature difference were above the appropriate range for the battery. After adopting PW/EG/CF CPCM temperature control, the battery temperatures at points 1 and 2 at the end of the 2C discharge were 43.86 °C and 43.13 °C, and the maximum temperature differences during 1C and 2C discharge were 1.59 °C and 1.33 °C, respectively. The temperature and temperature difference of the battery were in the appropriate range. The results show that the double-layer CPCM with SA–PA–LA/EG/CF and PW/EG/CF has evident advantages in temperature control. At the 1C discharge rate, only the inner layer CPCM is melted, and the temperatures at points 1 and 2 are 5.41 °C and 5.52 °C, respectively, lower than that of single-layer CPCM. This greatly reduces the temperature rise of the battery. Under the 2C discharge condition, the temperature control curve of the double-layer CPCM presents a double gradient phase transition, which can not only reduce the temperature rise of the battery, but also maintain the temperature of the battery in the lower temperature range of 30–40 °C for a long time. In comparison, the battery is primarily in the range of 40–45 °C when the temperature is controlled by a single-layer CPCM. Therefore, the arrangement of double-layer CPCM can better meet the temperature control requirements of the battery in all aspects (temperature, temperature rise, and temperature difference), under different discharge conditions. In addition, adding an aluminum honeycomb structure into the double-layer CPCM can further reduce the temperature and temperature difference of the battery, and the structure can also better meet the temperature control requirements during multiple charging and discharging cycles of a battery.

## Figures and Tables

**Figure 1 nanomaterials-11-01902-f001:**
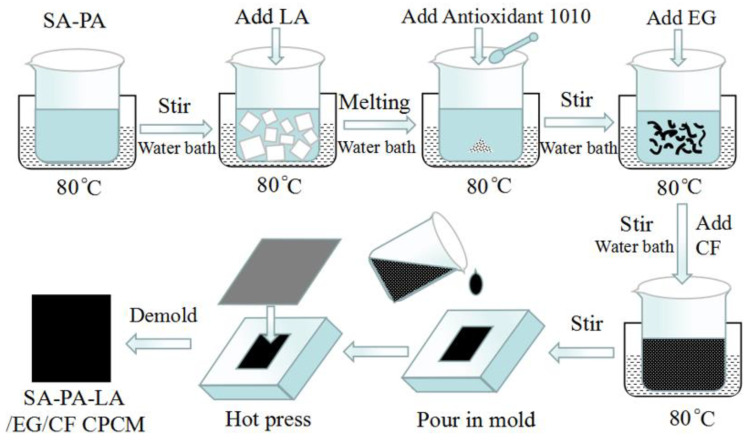
Preparation flowchart of SA–PA–LA/EG/CF CPCM.

**Figure 2 nanomaterials-11-01902-f002:**
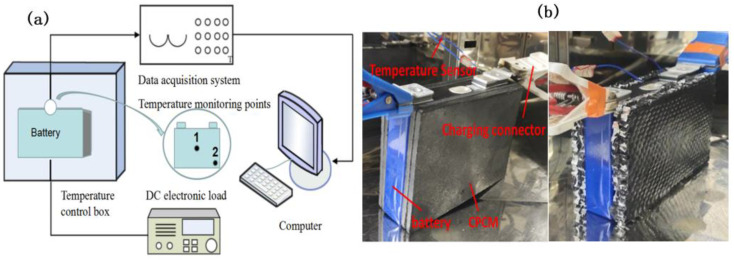
Battery charging and discharging experiment. (**a**) Schematic of battery charging and discharging system. (**b**) Battery heat dissipation module (double-layer CPCM structure without an aluminum honeycomb and double-layer CPCM structure with an aluminum honeycomb).

**Figure 3 nanomaterials-11-01902-f003:**
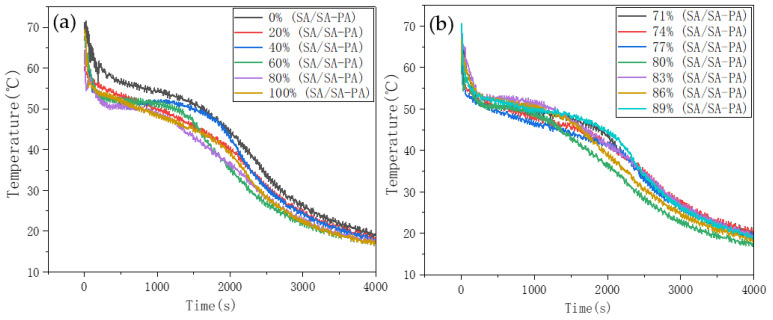
Step cooling curve of SA–PA binary PCM. (**a**) The first mass ratio of binary fatty acids. (**b**) The second mass ratio of binary fatty acids.

**Figure 4 nanomaterials-11-01902-f004:**
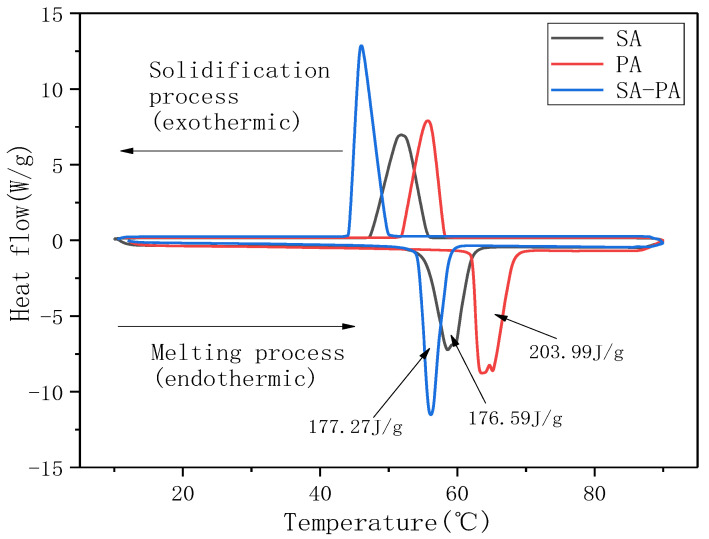
DSC curve.

**Figure 5 nanomaterials-11-01902-f005:**
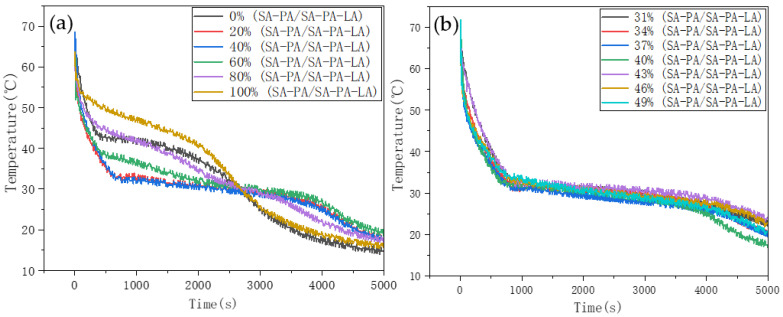
Step cooling curve of SA–PA–LA PCM. (**a**) The first mass ratio of ternary fatty acids. (**b**) The second mass ratio of ternary fatty acids.

**Figure 6 nanomaterials-11-01902-f006:**
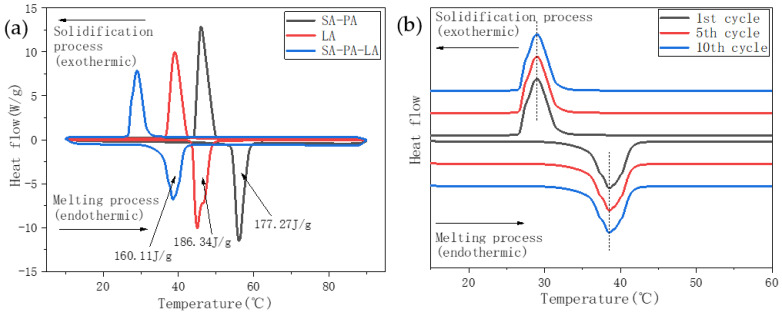
DSC curve. (**a**) DSC spectra of SA–PA, LA, and SA–PA–LA. (**b**) DSC cycle test spectra of SA–PA–LA ternary eutectic PCM.

**Figure 7 nanomaterials-11-01902-f007:**
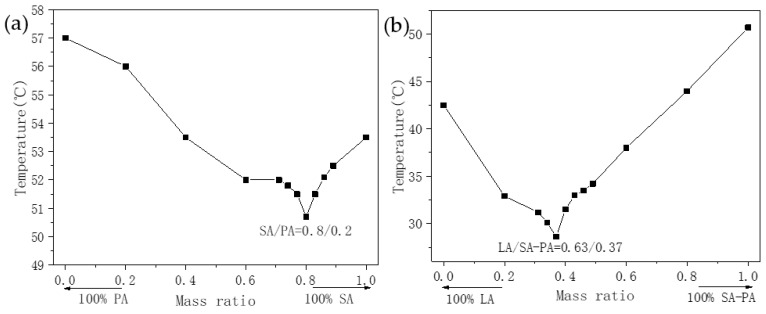
Equilibrium phase diagram of the eutectic systems. (**a**) SA–PA equilibrium phase diagram. (**b**) SA–PA–LA equilibrium phase diagram.

**Figure 8 nanomaterials-11-01902-f008:**
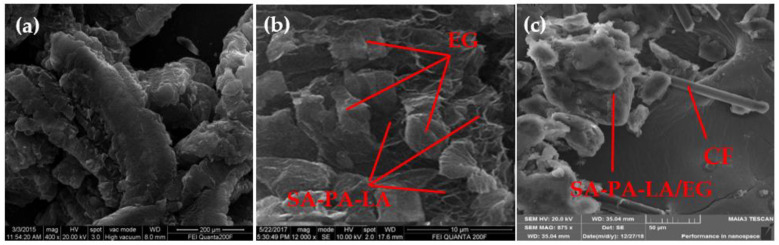
SEM images. (**a**) EG (**b**) SA–PA–LA/EG (**c**) SA–PA–LA/EG/CF.

**Figure 9 nanomaterials-11-01902-f009:**
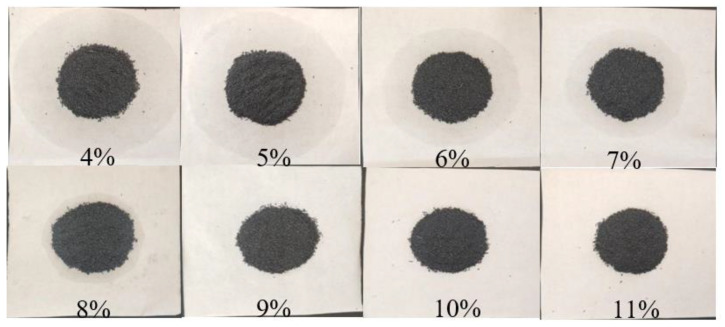
Leakage diagram of SA–PA–LA/EG CPCM with different EG contents.

**Figure 10 nanomaterials-11-01902-f010:**
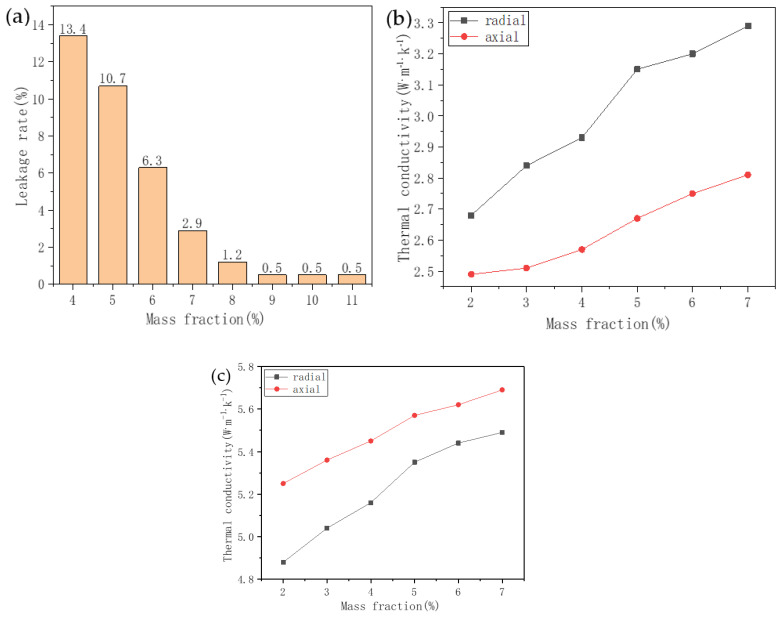
Leakage rate and thermal conductivity. (**a**) Leakage rate of SA–PA–LA/EG/CF CPCM with different EG contents. (**b**) Thermal conductivity of SA–PA–LA/EG/CF CPCM with different CF contents. (**c**) Thermal conductivity curve of SA–PA–LA/EG/CF CPCM with the aluminum honeycomb structure as a function of the CF content.

**Figure 11 nanomaterials-11-01902-f011:**
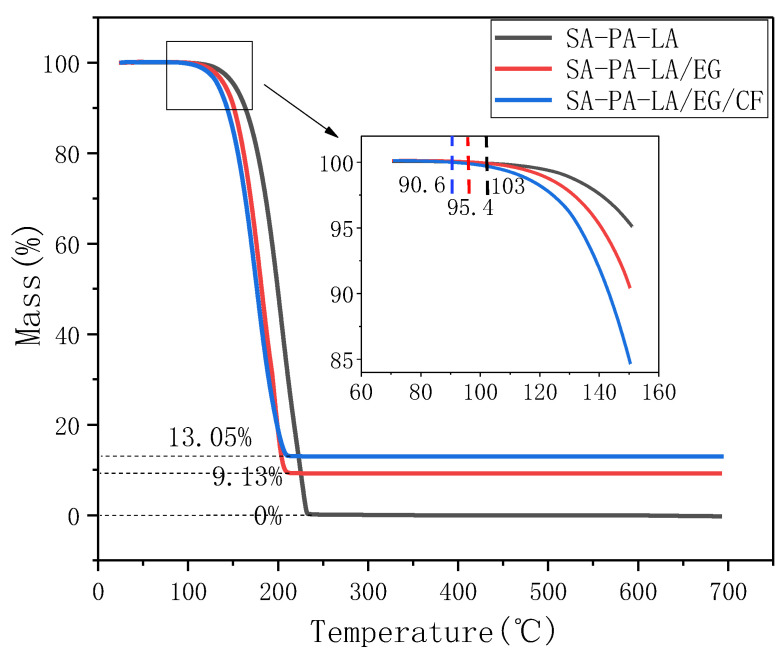
TGA curve.

**Figure 12 nanomaterials-11-01902-f012:**
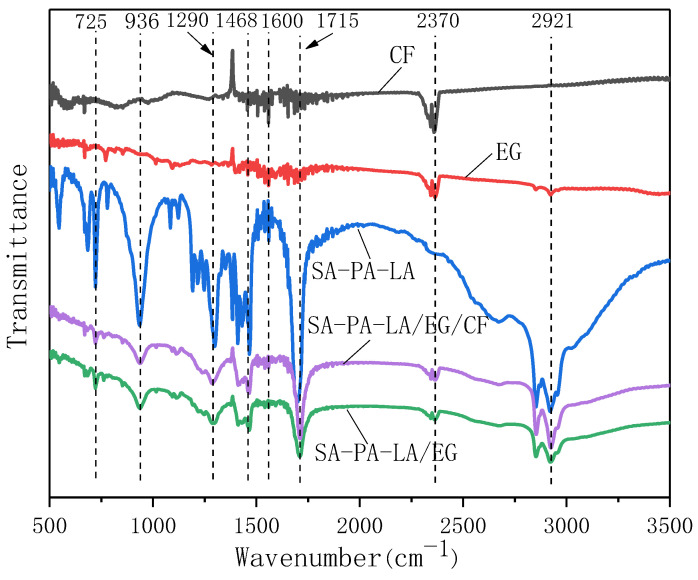
FTIR spectrum.

**Figure 13 nanomaterials-11-01902-f013:**
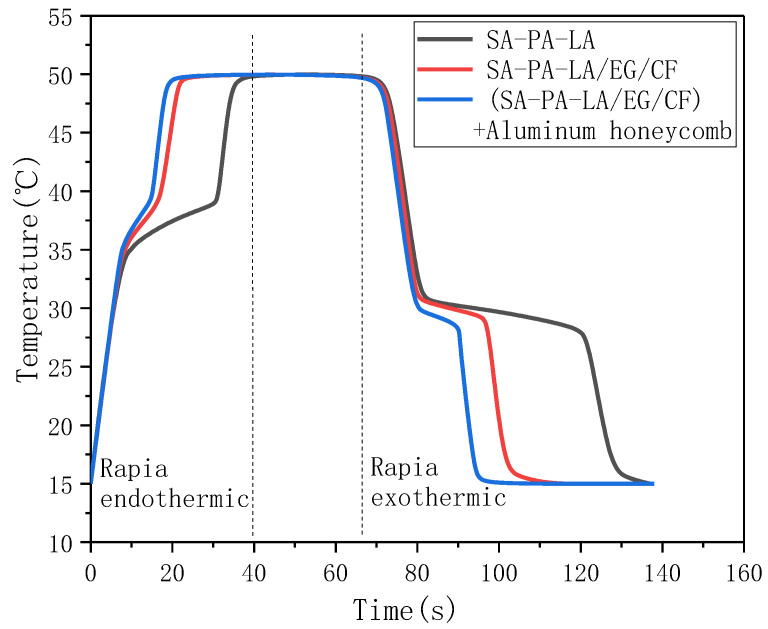
Temperature curve of heat storage and release.

**Figure 14 nanomaterials-11-01902-f014:**
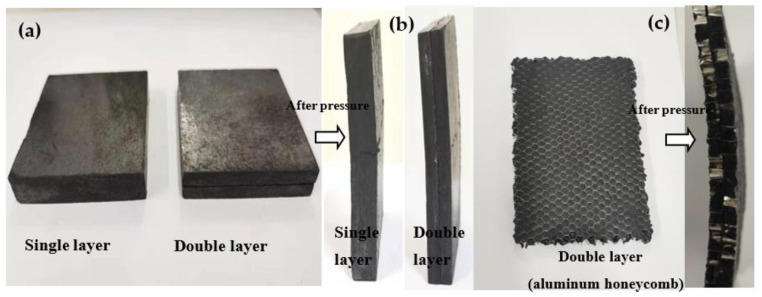
Stress and strain tests. (**a**) Physical image of the CPCM without aluminum honeycomb before compression. (**b**) Physical image of the CPCM without aluminum honeycomb after compression. (**c**) Physical image of the CPCM with aluminum honeycomb before and after compression. (**d**) Stress strain curve of the CPCM.

**Figure 15 nanomaterials-11-01902-f015:**
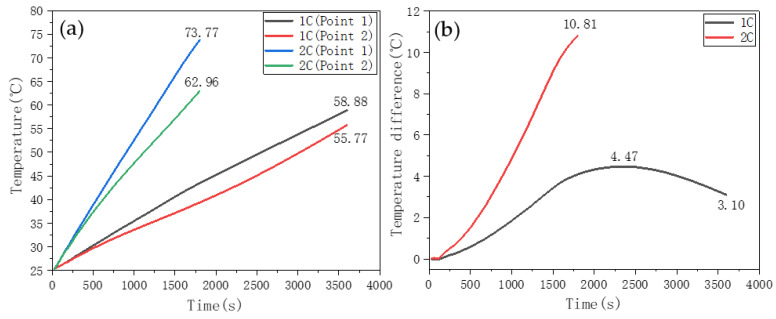
Battery discharge temperature rise curve without temperature control measures. (**a**) temperature change curve. (**b**) Temperature difference curve.

**Figure 16 nanomaterials-11-01902-f016:**
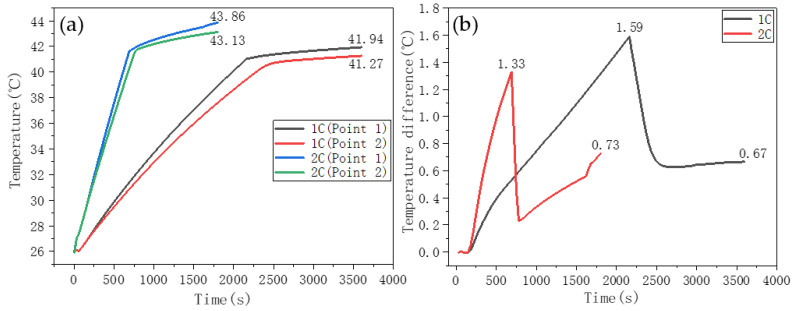
Battery discharge temperature rise curve using a single-layer CPCM to control the temperature. (**a**) temperature change curve. (**b**) Temperature difference curve.

**Figure 17 nanomaterials-11-01902-f017:**
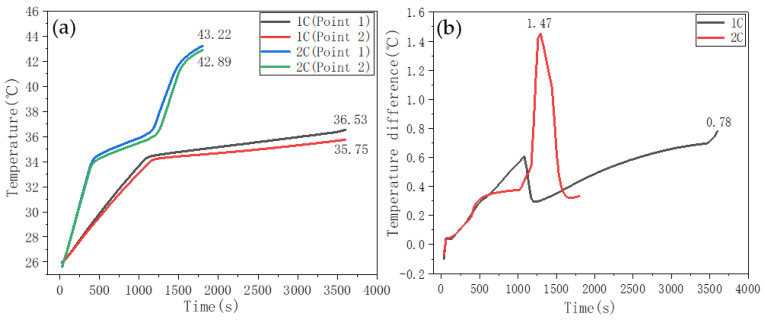
Battery discharge temperature rise curves using double-layer CPCM for temperature control. (**a**) temperature change curve. (**b**) Temperature difference curve.

**Figure 18 nanomaterials-11-01902-f018:**
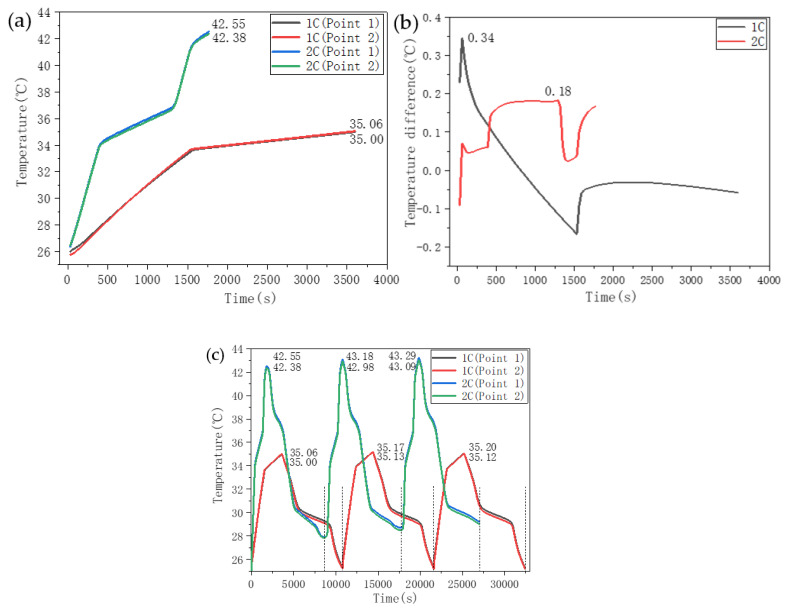
Temperature control curves of CPCM with a double aluminum honeycomb structure. (**a**) Temperature change curve. (**b**) Temperature difference curve between monitoring points. (**c**) Temperature curve of the charging and discharging cycles of the battery.

**Table 1 nanomaterials-11-01902-t001:** Battery parameters.

Parameters	Value
Thickness, width, height (mm)	28, 148, 98
Internal resistance (mΩ)	0.5
Capacity (Ah)	50
Weight (kg)	1.2
Charge and discharge cut-off voltage (V)	2.7–4.2
Nominal voltage (V)	3.7
Maximum charging current (C)	1
Maximum discharge current (C)	2

## Data Availability

Not applicable.

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
