# Peer review of "Preparation of SA–PA–LA/EG/CF CPCM and Its Application in Battery Thermal Management"

_nanomaterials, 2021, doi:10.3390/nano11081902_

Round 1

Reviewer 1 Report

This manuscript reports on preparation and application advantages of ternary low-melting phase change material (PCM), consisting of stearic acid, palmitic acid, and lauric acid. It is claimed good performance of this material in thermal management in batteries. Such ternaries are expected to be practical and stable enough for such purposes and are starting to attract much research interest.

The presentation in this paper is quite concise yet convincing, easy to read/perceive and discussion insights are adequately described and critically assessed.

Such work approaches the problem using the adequate methodology. It is timely, while especially useful because answering important questions (e.g., no chemical reactions between components in the ternary). The results are considered and discussed in non-trivial way. It is especially noteworthy the didactic way of presentation of the results and their interpretation in the right context.

In my opinion, the manuscript is bound to quickly attracting quite a significant interest also in relation to possible direct applications in batteries at larger scale.

There are excellent figures too.

There are some minor aspects/questions to this, otherwise excellent manuscript, that need attention; thus, it is acceptable for publication after a minor revision:  

1: Introduction is very well written. For the broadness of understanding of testing stability and possible chemical reactions in binary and ternaries of organic origin, it should be also mentioned that specially developed DFT theoretical approaches such as synthetic growth approach exist, are efficient, and have applied to complex systems [e.g., Journal of Physical Chemistry C 119 (2015) Pages 15735 - 15741; Dalton Transactions 44 (2015) Pages 3356 - 3366].

2: Thermal stability/decomposition issues (as per Sec. 3.2.3) should be discussed also in the context of bonding and chemistry of the components. The chemistry of the components is what defines the framework for the thermal stability/decomposition. Such discussion should also enhance the basic knowledge and understanding of the mechanisms behind thermal decomposition (when, and at the conditions, at which it happens).

3: In relation to the infrared spectrum analysis – it should be related to the theoretical works and relationship to structural/bonding aspects. The authors took the right direction referring to bonding, but it lacks systematics and reference to bonding as per theoretical works (many of them verified experimentally).

4: Spell-check and stylistic revision of the English of the paper are still necessary. Verbal forms and word order in many sentences can be improved.

Author Response

Dear reviewer:

Thank you very much for reviewing my manuscript! I have carefully read all your comments on my manuscript, and made a comprehensive revision of the manuscript based on your suggestions. In the revised manuscript, all the changes I made to the manuscript were tracked and marked using the revision mode. The specific amendments I made in the manuscript are as follows:

Point 1: Introduction is very well written. For the broadness of understanding of testing stability and possible chemical reactions in binary and ternaries of organic origin, it should be also mentioned that specially developed DFT theoretical approaches such as synthetic growth approach exist, are efficient, and have applied to complex systems [e.g., Journal of Physical Chemistry C 119 (2015) Pages 15735 - 15741; Dalton Transactions 44 (2015) Pages 3356 - 3366].

Response 1: I have carefully read the two research papers you recommended to me. I think the research methods mentioned in the papers are very novel and unique. Therefore, in Section 2.2 of the revised manuscript, I specially emphasized the effectiveness of the above methods, and cited these two papers in the References section (References [25] and [26]).

Point 2: Thermal stability/decomposition issues (as per Sec. 3.2.3) should be discussed also in the context of bonding and chemistry of the components. The chemistry of the components is what defines the framework for the thermal stability/decomposition. Such discussion should also enhance the basic knowledge and understanding of the mechanisms behind thermal decomposition (when, and at the conditions, at which it happens).

Response 2: According to your suggestion, from the perspective of material chemistry, I added specific analysis content of fatty acid and CPCM thermal stability mechanism in section 3.2.3 of the manuscript.

Point 3: In relation to the infrared spectrum analysis – it should be related to the theoretical works and relationship to structural/bonding aspects. The authors took the right direction referring to bonding, but it lacks systematics and reference to bonding as per theoretical works (many of them verified experimentally).

Response 3: According to your suggestions, I have added a systematic reference to theoretical knowledge in section 3.2.4 of this manuscript, and cited 3 related academic papers in the References section (References [30,31,32]).

Point 4: Spell-check and stylistic revision of the English of the paper are still necessary. Verbal forms and word order in many sentences can be improved.Spell-check and stylistic revision of the English of the paper are still necessary. Verbal forms and word order in many sentences can be improved.

Response 4: In this revised manuscript, I have revised the spelling and sentences of the full text, and invited the native English-speaking experts in related research fields to check the manuscript to ensure the accuracy and fluency of the sentences.

Reviewer 2 Report

The manuscript is interesting with valuable experimental data, and I support the publication.

I'd recommend to the authors to include a table with symbols/notations since there are many abbreviations that complicate the lecture, like 1C, 2C, etc.

Author Response

Dear reviewer:

Thank you very much for reviewing my manuscript! I carefully read your comments on my manuscript, and revised the manuscript based on your suggestions. In the revised manuscript, all the changes I made to the manuscript were tracked and marked using the revision mode. The specific amendments I made in the manuscript are as follows:

Point 1: The manuscript is interesting with valuable experimental data, and I support the publication.
Response 1: Thank you very much for your approval of my manuscript! In this revised manuscript, I have invited experts in related research fields of native English to further improve the manuscript, so as to ensure the accuracy and fluency of sentences and spelling.

Point 2: I'd recommend to the authors to include a table with symbols/notations since there are many abbreviations that complicate the lecture, like 1C, 2C, etc.

Response 2: According to your suggestion, I have added a table before the introduction of the manuscript to further clarify the meaning and full names of the abbreviations involved in this article.

This manuscript is a resubmission of an earlier submission. The following is a list of the peer review reports and author responses from that submission.

Round 1

Reviewer 1 Report

This manuscript reports on a composite phase change material based on expanded graphite-with prospects for applications in battery thermal management. The proposed manuscript contributes an original work featuring a well-chosen material system and its preparation/treatment that can be seen on one side as original and not much explored and on the other as a prototypical system. The authors succeed in a well-conceived and adequate characterization effort. In addition, the authors provide a good motivation behind adopting such a route for achieving this material system for thermal management. The analysis of the main results is presented with high degree of clarity. The presentation of the results is concise yet convincing, easy to read/perceive and discussion insights are adequately described and critically discussed.

Such work is really timely, while especially useful to the battery research community because of the didactic way the phase changes and the change of characteristics of the materials system were discussed as well as interpreted in the right context. There are other examples in the literature but by far insufficient yet.

In my opinion, the manuscript is bound to quickly attracting significant research interest also in relation to any ternary PCMs of similar nature.

There are excellent figures too.

There are some minor aspects/questions to this, otherwise excellent manuscript, that need attention; thus, it is acceptable for publication after a minor revision:

1: Title should be made clearer, avoiding the many abbreviations. As it is, it is very difficult to grasp/perceive even to specialized audience. It should be something like “Fatty acids treatment of composite phase change material based on expanded graphite with prospects for applications in battery thermal management”

2: Introduction is very well written. However, for the broadness of understanding of these and similar hybrid nanostructures, it should be also mentioned that theoretical methods such as focused DFT methods become increasingly available for addressing atomistic structure of such systems at this level of complexity thus helping rationalize thermal conductivity and other features [e.g., Journal of Physical Chemistry C 116 (2012) Pages 21124-21131 and Nanotechnology 27, (2016) Article number 055704]. They are good basis for addressing by simulations diversified material systems approached by reagents and should be paid attention to.

3: The motivation and discussion of exact choice of ranges of temperatures in the presented scheme are very important for such kind of study as the one reported in this manuscript. However, exactly this part of the text/discussion sounds a bit vague and telegraphic. Authors should elaborate on the concrete temperatures as related to melting points etc., --- in other words to elaborate on this subject a little bit more.

4: Generally speaking, it should be emphasized that the present approach and the discussion presented in this work may be applicable to other similar composite phase change nanomaterials.

5: Conclusions are nicely written but they may be improved by shortening them and focusing on the advantage to the applications. This will also make the final message of the paper clearer and help that the present work becomes better citable.

6: Spell-check and stylistic revision of the English of the paper are still necessary.

Reviewer 2 Report

In  this paper, the author investigated  the effect of the eutectic mass ratios  of component in the ternary low-melting phase change material on the heat dissipation efficiency of batteries. Based on the measurements of step cooling curves, DSC curves, and equilibrium phase diagram, they succeeded to determine the optimum ratio of SA-PA-LA. However, almost all of the results shown in this manuscript don't relate to nanomaterials. Therefore, I consider that this manuscript doesn't match the scope of this journal and the author should submit to other appropriate ones.

Reviewer 3 Report

Please see the document attached with my comments

Reviewer 4 Report

Title:

Preparation of SA-PA-LA/EG/CF CPCM and its application in battery thermal management

Comments:

The authors investigated the heat dissipation efficiency in batteries using CPCM materials during the battery usage. The topic is very important once the safety of the batteries should be improved. The work presents some drawbacks that will be further presented.

Point 1: The battery active material composition should be provided to better comparison with other works.

Point 2: The caption from several figures 15, 16 and 17 should verified.

Point 3: The efficiency of the heat dissipation should be provided for each study (CPCM single and double-layer).

According to that, this referee believes that this work should be reconsider after minor revision.